# A Dynamic Programmable Network for Large-Scale Scientific Data Transfer Using AmoebaNet

**Syed Asif Raza Shah [1]** and **Seo-Young Noh [2,\*]**

[1] Department of Computer Science, Sukkur Institute of Business Administration University (SIBAU), Sukkur, Sindh 65200, Pakistan; asif.shah@iba-suk.edu.pk

[2] Department of Computer Science, Chungbuk National University, Cheongjo 28644, Korea

\* Correspondence: rsyoung@cbnu.ac.kr

**Abstract:** Large scientific experimental facilities currently are generating a tremendous amount of data. In recent years, the significant growth of scientific data analysis has been observed across scientific research centers. Scientific experimental facilities are producing an unprecedented amount of data and facing new challenges to transfer the large data sets across multi continents. In particular, these days the data transfer is playing an important role in new scientific discoveries. The performance of distributed scientific environment is highly dependent on high-performance, adaptive, and robust network service infrastructures. To support large scale data transfer for extreme-scale distributed science, there is the need of high performance, scalable, end-to-end, and programmable networks that enable scientific applications to use the networks efficiently. We worked on the AmoebaNet solution to address the problems of a dynamic programmable network for bulk data transfer in extreme-scale distributed science environments. A major goal of the AmoebaNet project is to apply software-defined networking (SDN) technology to provide "Application-aware" network to facilitate bulk data transfer. We have prototyped AmoebaNet's SDN-enabled network service that allows application to dynamically program the networks at run-time for bulk data transfers. In this paper, we evaluated AmoebaNet solution with real world test cases and shown that how it efficiently and dynamically can use the networks for bulk data transfer in large-scale scientific environments.

**Keywords:** AmoebaNet; SDN; network as a service; bulk data transfer; QoS

## 1. Introduction

There has been a number of challenges emerged to large-scale data movement after the emergence of extreme-scale scientific applications. The data transmission is an essential function for scientific discoveries, particularly within big data environments. To date, there are several data transfer tools (such as GridFTP [1] and BBCP [2]) and services (such as the PhEDEx high-throughput data transfer management system, Globus Online [3], and the LIGO Data Replicator) have been developed to support the large-scale data transfer. These tools and services implemented the number of advanced data transfer features, such as partial transfer, transfer resumption, third-party transfer, and security. Although significant improvements have been made in large-scale data movement, but the existing data transferring tools are not up to the mark. These tools and services will require out of the box solutions to successfully meet the large-scale data transfer challenges.

The existing network paradigm that supports extreme-scale distributed science workflows consists of three major components: terabit networks that provide high network bandwidths, data transfer nodes (DTNs) and science DMZ architecture [4–6] that bypasses performance loopholes of scientific campus networks. On the other hand, Internet2 AL2S [7] and ESNet OSCARS [8] are mainly responsible

for on-demand secure path reservation over the Wide Area Network (WAN), and these solutions also provide the automated, guaranteed bandwidth services for scientific workflows. The existing network paradigm for scientific workflows has been proven very successfully. However, it has been observed during our comprehensive study that the present network paradigm for an extreme-scale scientific data transfer job still needs some more improvements to reach its full potentials. We claimed that the current networking paradigm must address the following major problems: scalability, last mile, and dynamic programmability problems.

The recent emerging concept in the network world is called Software-Defined Networking (SDN) [9–11]. This latest technology has been providing the new methods of configuration and management of networks. In SDN, the underlying network devices are simply considered as packets forwarding elements. It is possible to manage the control logic of the network centrally through a software program, which can effectively control the entire network behavior. In order to address such problems mentioned above, AmoebaNet [12] has been proposed.

It is common to notice low efficiency in data movement when running the data transfer tools on DTN machines. Such inefficient data transmission is one of critical points in scientific data computing. Therefore, it is required to provide high-performance, predictable, and schedulable data movement services for large-scale scientific facilities and their collaborators. Such challenges can be resolved by AmoebaNet, a Software-Defined Network (SDN) enabled service, which allows application to dynamically program the networks at run-time for bulk data transfers. AmoebaNet applies SDN technology to provide "QoS-guaranteed" network services in campus or local area networks. It is also possible that AmoebaNet complements existing network paradigm for extreme-scale distributed science: it allows applications to dynamically program the networks at run-time for optimum performance; and, it can be easily integrated with WAN circuit/path reservation systems such as ESNet OSCARS and Internet2 AL2S; Such capabilities of AmoebaNet can solve the problems of scalability, last mile, and dynamic programmability. In this paper, we particularly focused on an extension of AmoebaNet with some latest development of dynamic programmability. We evaluated AmoebaNet's feature in real world test cases along with a scientific application called BigData Express (BDE). The results show that AmoebaNet is able to enable the programmability feature efficiently as well as dynamically and to use the networks for bulk data transfer in large-scale scientific environments.

The rest of the paper is organized as follows. Section 2 discusses the background study related to our previously proposed solution AmoebaNet. A comprehensive study of a related work is discussed in Section 3. In Section 4, we discuss the evaluation and results of AmoebaNet's dynamic programmability feature with three real world test cases. Section 5 concludes the paper along with some future directions.

## 2. Proposed Solution

### 2.1. Overview of AmoebaNet Solution

We originally proposed AmoebaNet solution to support BigData Express (BDE) [13] scientific application. AmoebaNet project is currently ongoing evaluation and enhancement. There are several design goals for AmoebaNet: (1) our primary goal is "Network as a Service". This feature provides a dynamic programmable networking services for scientific applications to achieve the optimum performance level of networks. (2) QoS-guaranteed service. When applications require QoS guaranteed services for priority traffic, AmoebaNet must provide such types of services. (3) Wide applicability. AmoebaNet has to provide a mechanism for wide applicability to support a wide range of applications other than scientific applications only. (4) It must be easily deployable and operable. Followings are the salient features of AmoebaNet:

- To provide a dynamic programmability feature for underlying network resources, AMQP interface [14] was provided. The JSON-RPC style based communication system of AMQP will allow an application to interact with the underlying networks in a flexible manner.
- QoS-guaranteed path computation service along with route selection.

- Provide "Application-aware" networking services.
- Support differentiated service to ensure the required QoS services for priority traffic.
- Provision the QoS-guaranteed network path between two ends with a fastest possible time.
- REST-based network initialization and configuration.
- Path resiliency.
- Advanced reservation system.
- Service locking/unlocking.

As shown in Figure 1, the architecture of AmoebaNet consists of several internal services [12] that allow scientific applications (such as BDE) to program the underlying networks. AmoebaNet is an enterprise network service and we integrated and tested this service along with the ESNet OSCARS's wide area network connection service [15,16]. It can promptly provision end-to-end QoS guaranteed network paths across the multiple scientific domains. We implemented the AmoebaNet's service on top of the ONOS (Open Network Operating System) SDN controller [17,18] and developed the entire application in Java language. AmoebaNet successfully addressed the last mile, scalability, and dynamic programmability challenges for extreme-scale distributed scientific applications. In order to address these challenges, AmoebaNet introduced the following key features:

- By realizing network as a service, AmoebaNet supports various extreme-scale scientific applications and provides the rich and powerful set of dynamic programming primitives. These dynamic programming primitives make it possible for scientific applications to program the networks in more flexible ways. AmoebaNet also provides run-time programming capability for bulk data transfer, and utilizes the underlying network resources in an efficient and optimum way. These supported features of AmoebaNet address the agility, automation, programmability challenges.
- To address the last mile challenges, AmoebaNet provides a newly QoS-aware end-to-end bandwidth reservation feature. It is a new rate enforcement mechanism by combining the meters and queuing based techniques and characterizes the traffic into two classes of services: best effort and priority. AmoebaNet also has a feature of a new path computation service by using the largest bottleneck bandwidth algorithm [12]. In this mechanism the bandwidth can be considered as constraints and the best available paths between two end-points can be calculated. To solve the scalability challenges of science networks, AmoebaNet uses the spoke-hub distribution model. AmoebaNet at each site supports a hub-spoke distribution model for LAN or campus network. It allows seamless integration of local networks and wide area networks. When two systems are trying to send/receive the traffic to/from two physically isolated sites, then the service of AmoebaNet carefully identifies the corresponding p2p layer-2 circuits between sites and properly multiplexed and/or de-multiplexed the traffics accordingly.
- AmoebaNet is able to provide a novel control plane logic, which consists of different sub-modules. Each sub-module of control plane architecture is dedicated for a specific task and interacts with each other. The entire control plane architecture is implemented inside ONOS SDN controller as a service and it uses north-bound and south-bound APIs to interact with applications and data plane, respectively.
- To avoid the unsynchronized access and provides advanced reservation capability to network resources, we introduced the new locking/unlocking and scheduler mechanisms in AmoebaNet's solution.

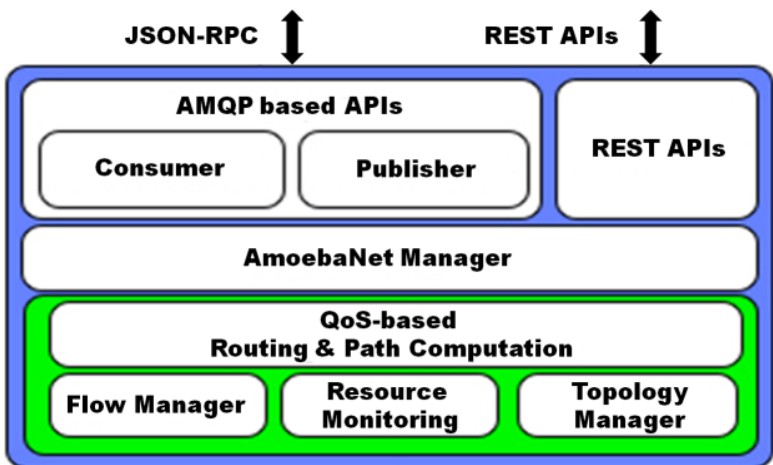

**Figure 1.** Architecture of the AmoebaNet solution.

The more detailed information about AmoebaNet can be found in [12] for its design, implementation, and evaluation.

### 2.2. AmoebaNet's Programming Primitives

One of the key features of AmoebaNet is to achieve dynamic programmability and to make it possible for scientific applications to program networks dynamically. In order to implement such a feature, AmoebaNet provides different programming primitives, which can provide applications that realize network as a service (NaaS). The list of programming primitives supported by AmoebaNet is shown below:

- initial_config(): By using initial configuration programming primitive, an application can configure the gateway switches' interfaces/ports. The gateway switches are connecting the internal campus networks with the external networks. To provision the layer 2 end-to-end network paths required switching of different path segments at gateways. The initial configuration will help to setup VLANs and perform operations such as push, pop, and swap VLAN tags from internal to external network and vice versa. An example of the layer 2 end-to-end network path is shown in Figure 2, gateway switches (A and B) are configured for two sites (A and B), respectively. In fact, the path segments is made of different VLANs (VLAN1 ↔ VLAN2 and VLAN2 ↔ VLAN3).
- lock()/unlock(): It is a new feature introduced in our proposed solution, which is called lock/unlock. As per best of our knowledge, no API of ONOS is found, which is providing the same functional features to applications. AmoebaNet provides the lock/unlock mechanism to prevent the resource contention and unsynchronized access of resources among the applications. When an application sends the query request for available resources, then the entire session will be locked for that request. It keeps the lock until reservation requests are committed or the default session time expires (i.e., five seconds). Some applications (like BigData Express) require to hold the resources for a while for coordination purpose and others have not such type of requirements. Therefore, it can be claimed that AmoebaNet supports wide spectrum of applications; it provides the flexible usage of lock/unlock mechanism as per the requirements of applications. Simply, an application can enable or disable the lock/unlock during the query request. At this moment, AmoebaNet supports a very high-level coarse-grain lock/unlock, but our target is to provide the fine-grain for lock/unlock in future.
- query_virtual_netslice(): The current intent framework of ONOS can provide a functionality to install multi-to-single host or single-to-multi host paths. However, this intent framework is not providing the functionality of virtual slicing based on VLANs along with QoS-guaranteed features.

In our proposed AmoebaNet solution, we added this feature by using virtual network slicing programming primitives.

- query_path(): By using the query path programming primitive, an application can view the current setting of a particular path;
- query_host_to_host(): This programming primitive helps to view the available maximum bandwidth between two DTNs in a network segment;
- query_host_to_gateway(): To check the available bandwidth from a host to a gateway switch;
- create_path(): Using this programming primitive an application can dynamically establish a layer 2/3 path between two end points. It is also necessary to specify the type of traffic and required amount of bandwidth for a specific time period;
- create_virtual_netslice(): Given a list of IPs, it creates or reserves a virtual network slice among them with a specified bandwidth and at a specified time slot;
- update_path(): Used to change the settings of installed end-to-end path; this programming primitive will seamlessly update or modify the paths information;
- update_virtual_netslice(): Used to change the settings of installed virtual slice path; it seamlessly updates or modifies the slice paths information;
- release_path(): Torn down and deleted the entries of a particular path;
- release_virtual_net_slice(): Torn down and deleted the entries of a particular virtual network slice.

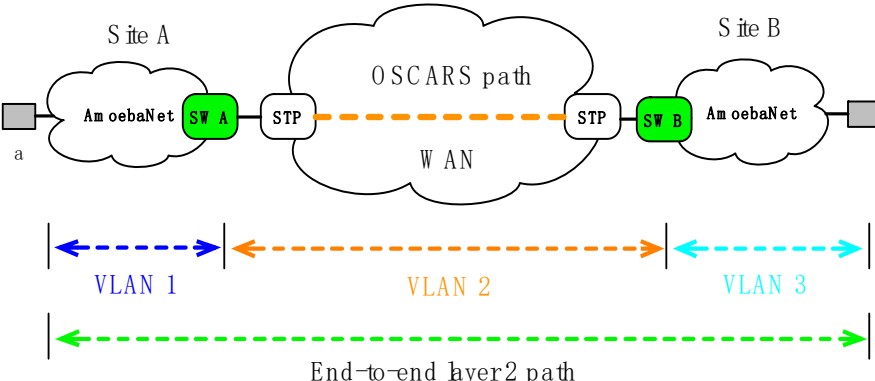

**Figure 2.** VLAN operations at Gateways.

### 2.3. AmoebaNet Solution for the BigData Express (BDE) Project

AmoebaNet is originally designed for BigData Express application. A major goal of BigData Express is to apply AmoebaNet's networking service to provide "Application-aware" networks and facilitate large-scale data transfer. BigData Express has been designed for large scientific computing centers. It has been deployed in many government supported computing centers such as Fermi National Accelerator Laboratory supported by the US Department of Energy. Figure 3 is illustrating two typical sites connected by BigData Express. These sites consist of high-performance DTNs using a dedicated cluster, a large-scale storage system, and an SDN-based AmoebaNet campus networking service in BigData Express. An on-demand WAN-based service requires that BigData Express establishes the site-to-site connection. It also enables guaranteed bandwidth by creating paths between source and destination sites in designated time slots.

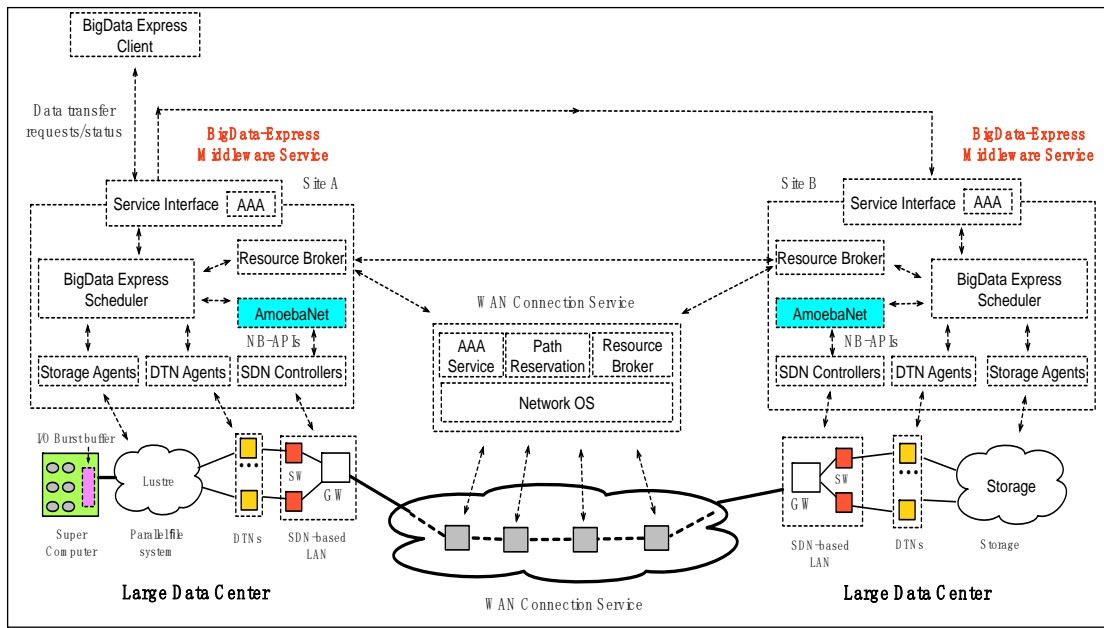

**Figure 3.** AmoebaNet service along with BigData express architecture.

The scheduler of BigData Express (BDE) is logically centralized and mainly responsible for programming the entire networks dynamically and performs data transfer jobs. The purpose of the BDE scheduler is to coordinate with each BigData Express site to properly program the underlying networks. It is also responsible for scheduling and managing the local site's resources such as storage, DTNs, and the AmoebaNet for campus networks through agents (storage agents, DTN agents and AmoebaNet). AmoebaNet will keep track of the BigData Express LAN topology and traffic status with the aid of SDN controllers. As requested by the BigData Express scheduler, it will program a network at run-time for data transfer tasks.

## 3. Related Work

The main goal of this research work was to allow scientific applications to dynamically program the networks for bulk data transfer. It provided high performance, scalable, end-to-end, and programmable networks and enables scientific applications to use the network most efficiently. We utilized the advanced features of SDN technology for the capability of dynamic program [19,20] networks and provided end-to-end QoS-guaranteed network service for scientific applications. In recent, there are a number of studies that have proposed similar goals. In this section, we reviewed the related work and provided the key differences between previous works and our proposed solution.

A dynamic flow scheduling system, called Hedera, was proposed by in [21]. Hedera can reduce traffic collisions by using the adaptive scheduled switching fabric technique. Such a solution has been adopted in the PortLand testbed [22]. It is claimed that Hedera is able to manage the modest control and computation overheads, and more over increases the overall bandwidth up to four times higher than the conventional ECMP technique. The core design technology proposed in Hedera was based on a multi-root network topology tree, which uses the large number of parallel path between two edge switches. However, Hedera network architecture is not adoptable for Science DMZ networks because of its multi-root tree network, which creates the overhead of a large number of parallel path between two ends.

Wang et al. [23] achieved the considerably lower processing overhead by installing the wildcard packet handling rules using OpenFlow. Such an approach uses the technique of load balancing at each replica of servers. There are several approached can been found in [24] to reduce the overall

cost associated with the OpenFlow's fine-grained control. In [25], research work helps to improve SDN scalability.

To achieve the current traffic demands, SWAN [26] proposes a novel link utilization technique for inter data center networking. In this research, the author employed SDN technology to a re-configuration of data plane to match the traffic needs of applications. Google's B4 [27] proposed a similar type of solution, which addresses the problem of traffic engineering. To achieve the demands of applications, B4 splits the traffic flows into multiple paths to balance the load of applications as per given priorities. A single domain design architecture has been proposed in both B4 and SWAN solutions. Both are using a centralized SDN controller server to orchestrate all activities. However, the multi-domain problems have not been addressed in B4 and SWAN, which is an essential demand of scientific networks. In contrast, our proposed AmoebaNet solution complements the current network architecture (i.e., ESNet OSCARS) and provides complete integration services to support large scale data transfer services for multi-domain scientific centers.

SDN security is one of hottest research topics. There are many research works for SDN security issues [28–31]. Such security was out of our research scope, which is expected to be covered in our future work.

The recent invention in SDN technology is P4 programming language, which has been proposed by the founder of SDN [32]. P4 is different from OpenFlow protocol; it has been designed to program and control the pipeline level of the data plane of SDN enabled routers and switches. The advent of P4 programming language introduced the new form of flexibility for users to control the network with a top-down approach [33–36].

To provide the high performance end-to-end networking service for extreme-scale distributed science, we need to combine the concepts of DTNs and Science DMZs, emerging software-defined networking technologies and terabit wide area networks. The pacific research platform (PRP) [37] is an example of providing such type of network environments. It has been designed and implemented with the support of the Corporation for Education Network Initiatives in California (CENIC). The core concept of PRP was to provide a collaborative research network particularly for multi-domain large scientific centers. With the help of the National Research Platform (NRP), US National Science Foundation (NSF) is planning to expand the idea of PRP to the national scale [37]. The current initiatives of NSF are only limited to the physical infrastructure level, which supports the extreme scale data transfer across multiple-domains, however, the middleware software and services are still missing for large-scale optimized data movement. To support large-scale data movement operations, the end-to-end QoS-guaranteed network paths are essential requirements for scientific applications. The fast provisioning along with the optimized QoS-guaranteed network path is still a long sought mission for scientific communities.

In this paper, we proposed the extension of AmoebaNet solution, which can cover the middleware elements for extreme-scale scientific data movement. Our solution implements the SDN enabled dynamic network service that is capable of creating end-to-end QoS-guaranteed network path for large-scale scientific centers. In addition, it has the capability to dynamically program the local networks in a distributed and coordinated way for data transfer services and fulfill the requirements of scientific applications.

A dynamic provisioning of layer 2 circuits for multi-domain architecture is proposed by GEANT, which is known as the AutoBAHN [38] provisioning tool. The main focus of the AutoBAHN tool is to enable interoperability across multiple domains. When different domains are deployed with different network technologies, then AutoBAHN can easily negotiate and establish QoS-guaranteed network path. However, the major limitations of AutoBAHN are application layer support and layer 3 circuit support. It can provide better services for WAN deployment, which is not considered as end-to-end QoS-guaranteed network path for scientific applications. Finally, AutoBAHN is yet not deployed and implemented in real testbeds.

To achieve end-to-end layer 2 QoS-guaranteed service for scientific data movement, DYNES (dynamic network systems) project were initiated in [39], and it was deployed at local area networks using the ESNet OSCARS service. DYNES framework combines the OSCARS enabled LAN and WAN services at campus networks. However, such types of frameworks are not appropriate and successful at campus networks because OSCARS is not the suitable solutions for LANs.

A NSF-funded research project launched with the name of DANCES (Developing Applications with Networking Capabilities via End-to-end SDN) [40]. DANCES uses SDN technology. It enables the bandwidth scheduling for networks and its aim is to achieve the maximum performance of scientific applications. To schedule and manage the bandwidth for applications, DANCES proposed a Centralized OpenFlow and Network Governing Authority (CONGA) service. It was basically used to schedule networking resources efficiently. Due to the centralized architecture approach, CONGA in DANCES raised the problems of scalability and limiting the acceptability for extreme-scale scientific applications.

It is also important to consider small file data transfer over the wide area and how to reduce the per-file overhead. In [41] a comprehensive solution was proposed by using concurrency and prefetching techniques. On the other hand, a systematic examination of large-scale data transfer presented in [42], which gives us a complete analysis of very small and large-scale datasets. However, in our case, to deal optimally with small and large-scale datasets during data transfer is the main responsibility of top tier scientific application (such as BigData Express).

## 4. Evaluation and Results

In this section we evaluated the AmoebaNet service with three different bulk data transfer test cases. Those cases were performed with a BigData Express scientific application.

### 4.1. Test Case I: Extreme-Scale Data Transfer in a Single Domain

To prove the ability of higher throughput with higher bandwidth, we designed a real experiment of extreme-scale data transfer by using AmoebaNet in a single domain. As illustrated in Figure 4, this experiment's topology consisted of two physical SDN-enabled switches (Pica8 P3930 and P5101) and both switches were connected back-to-back with each other by using the high speed link of 40GE. We also connected four data transfer nodes (DTNs) in topology with 40GE interfaces of switches and each DTN was configured with virtual interfaces (VLANs).

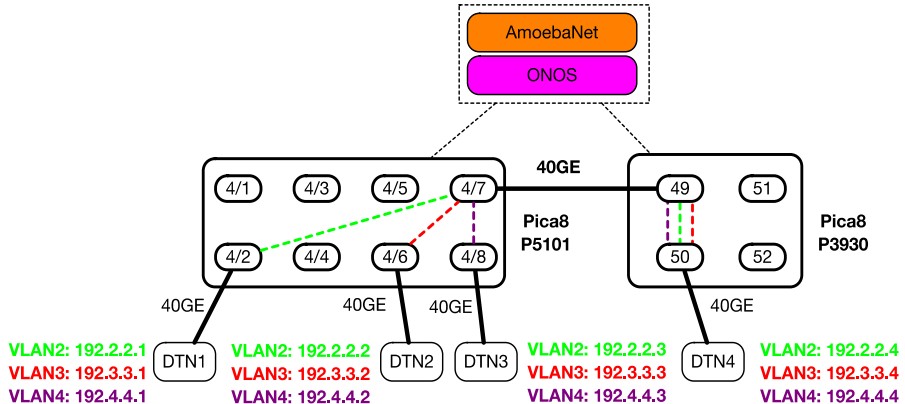

**Figure 4.** Extreme-scale data transfer topology in a single domain.

We performed large-scale data transfer jobs between data transfer nodes by using the iperf tool and classified the traffic into two different classes (priority and best effort). We included three types of large-scale data transfer jobs in our experiment, which lasted for 20 min. Following are the details of each job:

- Priority traffic 1, which is sent from 192.3.3.2 (vlan3 nic@DTN2) to 192.3.3.4 (vlan3 nic@DTN4). The data transfer starts at 4th min, and ends at 15th min, wherein the maximum bandwidth is capped at 10 Gbps.
- Priority traffic 2, which sent from 192.4.4.3 (vlan4 nic@DTN3) to 192.4.4.4 (vlan4 @DTN4). The data transfer starts at the 7th min and ends at the 18th min, wherein the maximum bandwidth is capped at 15 Gbps.
- Best-effort traffic, which is sent from 192.2.2.1 (vlan2 nic@DTN1) to 192.2.2.4 (vlan2 nic@DTN4). The data transfer starts at the 1st min and ends at the 20th min.

Initially we established the end-to-end paths each data transfer job and used a well-defined simple application to dynamically program the network resources using AmoebaNet. Essentially, AmoebaNet performs these tasks:

(1)    Calculate and route local LAN paths.
(2)    Install flow rules to set up local LAN paths.
(3)    Install meters and queues to enforce rate control and QoS guarantee for priority traffic.

Following is an example of command to establish a priority path using AmoebaNet:

```
{
  "cmd":"create_path",
  "hosts":
    [{
"startTime": "immediate",
"endTime": "2019-09-07 13:00",
      "srcIp":"192.3.3.2",
      "rate":"10000",  // in Mbps
      "dstIp":"192.3.3.4",
      "trafficType":"priority traffic",
      "vlanId":"3",
"routeType": "host-to-host"
    }]
}
```

We collected throughput measurements for the data transfer jobs. The evaluation results are illustrated in Figure 5. As shown, (a) from 1 to 4 min, best-effort traffic consumed the entire 40 Gbps bandwidth of the 192.2.2.1—192.2.2.4 path. (b) From time 4 to 7 min, priority traffic 1 achieved a capped throughput of ~10 Gbps, with the best effort traffic consuming the remaining ~30 Gbps. (c) For time 7–15 min, priority traffic 2 was sent and attained a throughput of ~15 Gbps, which reduced the best-effort traffic throughput to ~15 Gbps, and priority traffic 1 remained unchanged at ~10 Gbps. (d) Between time 15 and 20 min, both priority traffic 1 and 2 transfers were completed, allowing the best-effort traffic's throughput to increase to ~40 Gpbs. The best-effort traffic consumed the entire bandwidth of the end-to-end path.

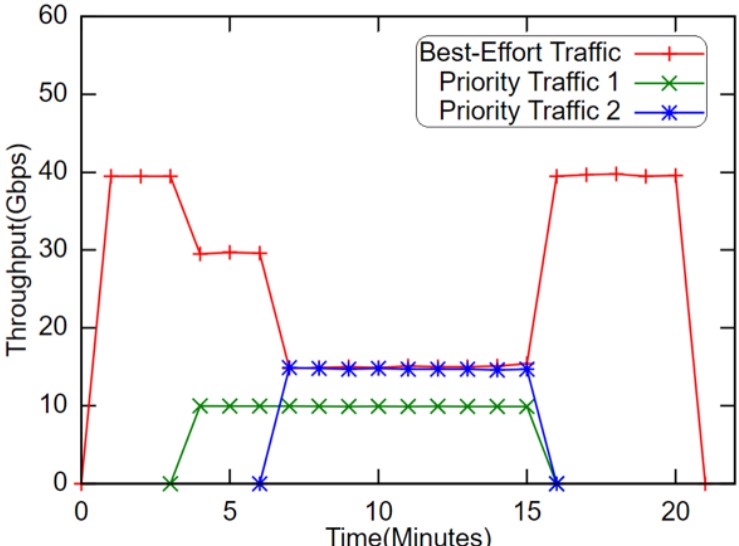

**Figure 5.** Extreme-scale evaluation results.

Figure 5 clearly demonstrates that AmoebaNet can work for the extreme-scale data transfer job, and it allows application to dynamically program the underlying networks.

### 4.2. Test Case II: Cross Domain Data Transfer

Network topology of our first test case is illustrated in Figure 6. This test case consisted of three 40GE DTNs—BDE1, BDE2, and BDE3. We connected the DTNs to an SDN-enabled network switch (Pica8), and also configured the single or multiple VLANs interfaces on each DTN server. We divided our test case into two logical sites—A and B. Both sites A and B are highlighted in blue and green, respectively. At our SDN-enabled network switch, we configured the port 4/1 as a gateway for site A and port 4/5 as a gateway for site B. To connect both sites (A and B), we simulated a WAN networking by using OSCARS layer 2 circuit to establish the loopback path at the boarder router of each site. The ESNet OSCARS service was used to dynamically set up this loopback path between two sites. The total bandwidth of the loopback path was 1 Gbps and round trip time was approximately 92 ms.

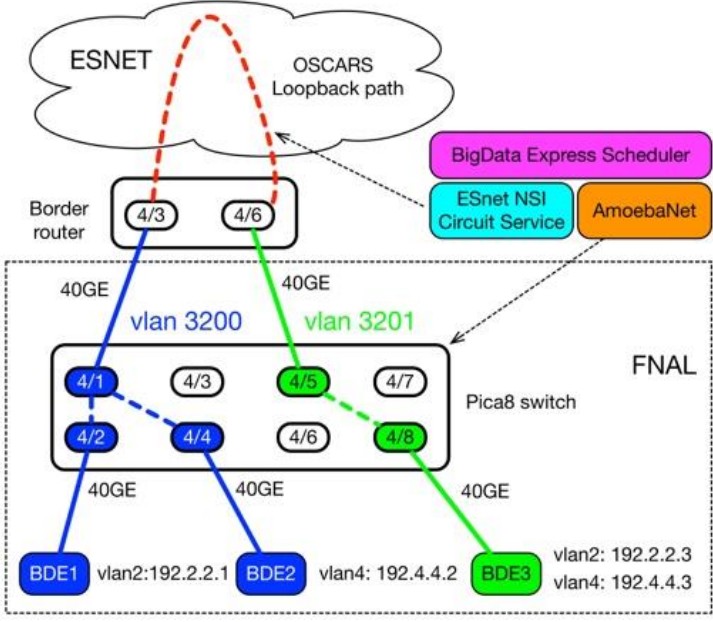

**Figure 6.** Cross domain data transfer test case.

In our evaluation process, we performed the basic data transfer experiments between two sites A and B with the help of a well-known iperf tool, and we used two different classes of traffic in this test case:

- Priority traffic: In this case, we sent the data from BDE1 data transfer node (192.2.2.1 with vlan2) to BDE3 data transfer node (192.2.2.3 with vlan2). The traffic started at 120 s, and ended at 720 s, wherein the maximum bandwidth was capped at 700 Mbps.
- Best-effort traffic: In this case, data was sent from BDE2 data transfer node (192.4.4.2 with vlan4) to BDE3 data transfer node (192.4.4.3 with vlan4). The traffic started at 0 s, and ended at 600 s.

Before data transfer started, the BigData Express scheduler established end-to-end paths between DTNs. First, it called the ESNet OSCARS service to establish the loopback WAN path. Then, we dynamically programmed the network at run-time by using the AMQP JSON-RPC primitives of AmoebaNet services. In this experiment, the following tasks were performed by the AmoebaNet service: (a) the campus network's paths calculation and route selection; (b) identified the gateway ports and OSCARS's VLANs IDs at both sites A and B and installed the flow rules for each campus network path; it used 3200/3201 VLAN IDs to connect a campus network with the ESNet OSCARS layer-2 WAN path; and (c) it enforced guaranteed QoS services for priority traffic by dynamically installing the queues and meters.

Figure 7 shows the evaluation results. It shows that: (a) from 0 s to 120 s, best-effort traffic had a throughput close to 1 Gpbs, which implies that it occupied all the bandwidth of the OSCARS WAN path, and (b) from 120 s to 600 s, priority traffic had a throughput of ~700 Mbps while best-effort traffic's throughput decreased to ~300 Mbps. This is because priority traffic had a higher priority. When priority traffic started at 120 s, it would occupy the bandwidth as much as possible until it was capped at 700 Mbps. Best-effort traffic then took the residual bandwidth, which was ~300 Mbps. (c) From 600 s to 720 s, best-effort traffic stopped. Although the total available bandwidth was ~1 Gpbs, priority traffic was still capped at 700 Mpbs.

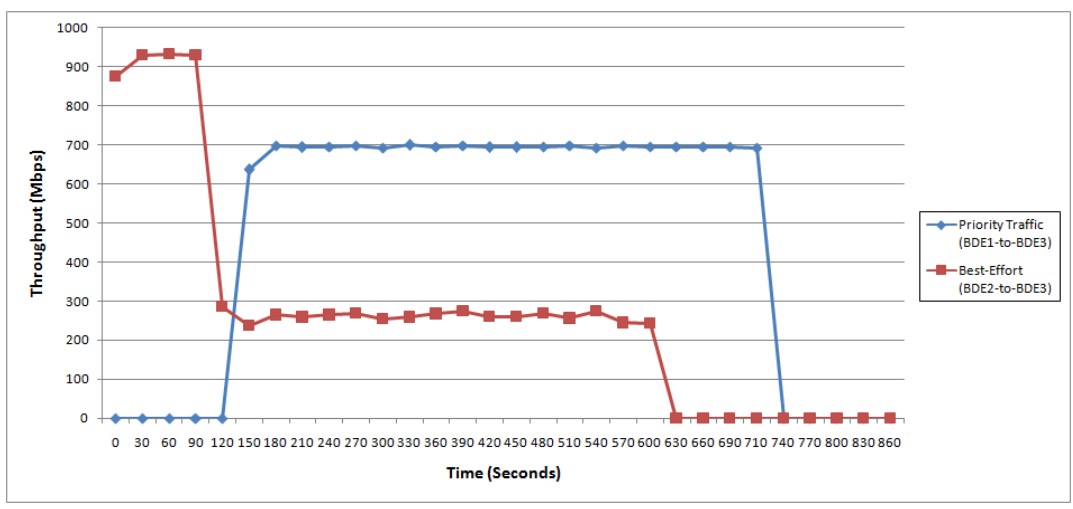

**Figure 7.** AmoebaNet evaluation results.

In this test case, it was clearly demonstrated that the AmoebaNet provided a mechanism for applications to dynamically program the networks at run-time for bulk data transfer jobs, and it could provide "Application-aware" networking services as well. AmoebaNet was also able to support differentiated service to ensure the required QoS services for priority traffic. It is also important to note that AmoebaNet had the ability to provision the QoS-guaranteed network paths between two ends with the fastest possible time and provided the enterprise level network services along with the ESNet OSCARS WAN connection.

*4.3. Test Case III: Cross Pacific QoS-Guaranteed Data Transfer*

In this test case, we evaluated AmoebaNet in a cross pacific domain along with BigData Express scientific application to demonstrate its major functions, capabilities, and features. AmoebaNet provided APIs to BDE to program the underlying campus network for provisioning on-demand end-to-end QoS guaranteed paths.

As shown in Figure 8, our experimental topology consisted of two independently managed scientific computing facilities—the KISTI site and Fermi National Accelerator Laboratory (FNAL) site—which was connecting these sites with a layer-2 dedicated WAN path. In particular, the ESNET circuit segment could be dynamically set up and torn down using the ESNET NSI circuit service. The maximum available end-to-end bandwidth between FNAL and KISTI was 10 Gbps. In this topology, the FNAL site was logically divided into two different sites which consist of the separate AmoebaNet-based SDN controllers, SDN switches, and DTNs. Sites configurations are as shown below:

- FNAL sites:

  ○　Site 1: Fermi National Accelerator Laboratory (FNAL):

  - DTNs: DTN1 (BDE1), DTN2 (BDE2), and DTN3 (BDE3).

    - Data transfer nodes equipped with: a 40GE Mellanox NIC and an Intel NVMe storage drive.
  - Switches: one SDN enabled switch (Pica8 P5101 running PicaOS).
  - ONOS SDN controller along with AmoebaNet solution.

  ○　Site 2: Fermi National Accelerator Laboratory (FNAL S2):

  - DTNs: DTN4 (BDE4), and DTN5 (BDE-HP5).

    - Data transfer nodes equipped with: a 40GE Mellanox NIC and an Intel NVMe storage drive.
  - Switches: one SDN enabled switch (Pica8 P3930 running PicaOS).
  - ONOS SDN controller along with AmoebaNet solution.

- KISTI site:

  ○　DTNs: DTN2, DTN3, and DTN4.

  - Data transfer nodes equipped with: a 10GE NIC and an Intel NVMe storage drive.
  ○　Switches: one SDN enabled switch (Z91000 HP running PicaOS).
  ○　ONOS SDN controller along with AmoebaNet solution.

To evaluate AmoebaNet functionality, we configured AmoebaNet and all related software of BDE at FNAL and KISTI sites. The data transfer service is accessible for users using BDE web portal: https://yosemite.fnal.gov:5000 (accessible URL at FNAL site), or https://134.75.125.77:2888/ (accessible URL at KISTI site), respectively. In the evaluation, two parallel data transfer tasks were submitted at https://134.75.125.77:2888 (BDE web portal at KISTI):

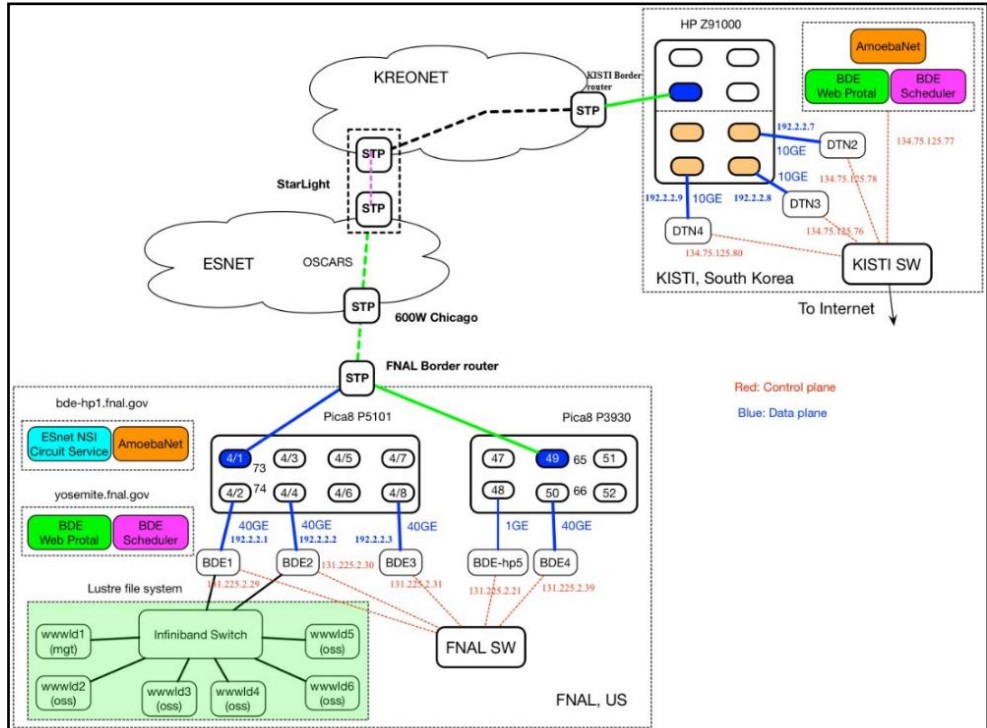

**Figure 8.** AmoebaNet's cross Pacific experimental topology.

- Task 1: We setup 1 Gpbs dedicated bandwidth for QoS-guaranteed end-to-end path and transferred a 372.5 GB data set from DTN2 at KISTI to BDE1 at the first logical site of Fermilab (i.e., FNAL).

- Task 2: We configured 1 Gbps dedicated bandwidth for QoS-guaranteed end-to-end path and transferred a 20 GB data set from DTN2 at KISTI to BDE3 at the first logical site of Fermilab (i.e., FNAL).

Both tasks were submitted in parallel and both tasks started a data transfer job at the same time. When both tasks were submitted using BDE web portal, the BDE software immediately started communication with AmoebaNet and sent queries for available bandwidth of each local sites. Once the available bandwidth satisfied the requirements, then BDE software sent a JSON formatted command using APIs to AmoebaNet for reservation of bandwidth at both sites.

The evaluation results of the parallel submitted tasks are shown in Figure 9. Task 1 successfully completed its data transfer job from KISTI (DTN1) to FNAL (BDE1) at the transfer rate of approximately ~130 MB/s, which is equal to ~1 Gbps. On the other hand, task 2 also completed its jobs from KISTI (DTN1) to FNAL S2 (DTN4) and the data transfer rate was approximately ~130 MB/s, which is also equal to ~1 Gbps.

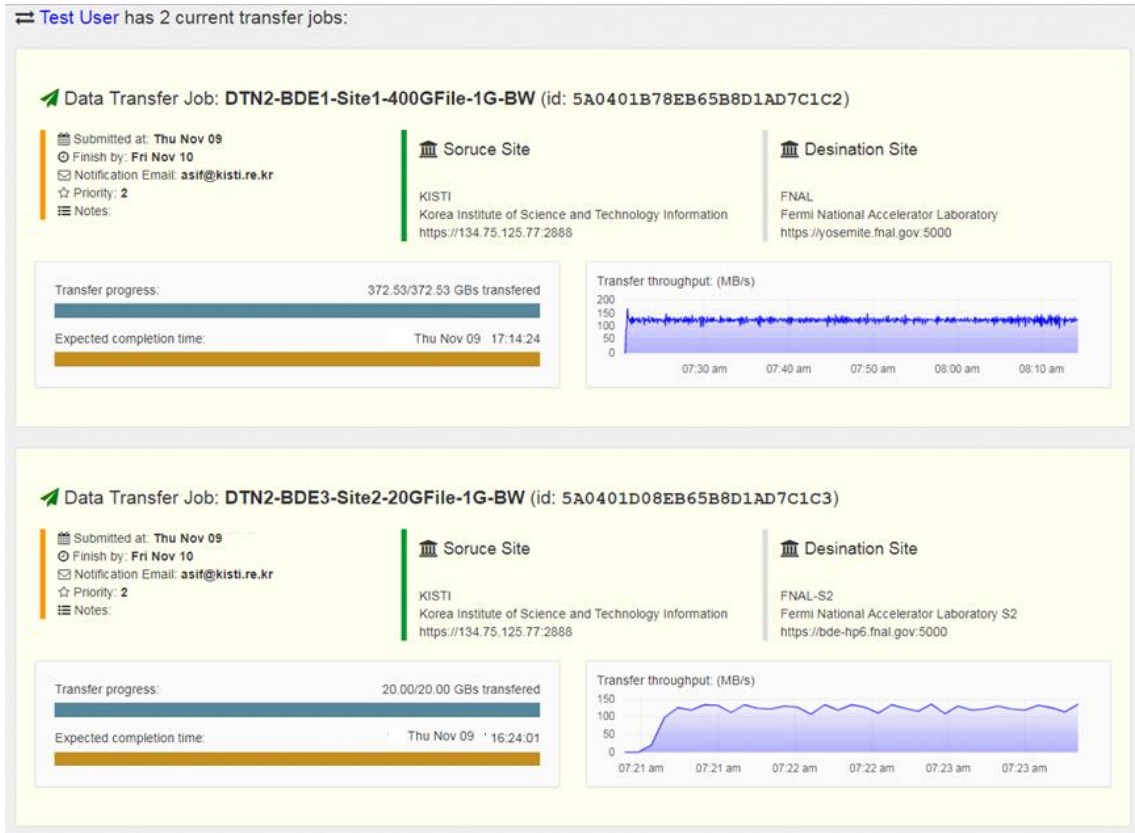

**Figure 9.** Evaluation results of the parallel submitted tasks.

## 5. Conclusions

In this paper we have shown how to program the network at run time for bulk data transfer jobs using AmoebaNet and BigData Express application. To achieve the QoS-guaranteed network services for scientific campus networks, we applied SDN technology for AmoebaNet. In this paper we tested the AmoebaNet solution with three different test cases to achieve the goal of a dynamic programmable network for large-scale scientific data transfer. The evaluation of test cases clearly showed that the AmoebaNet allowed applications to dynamically program the networks at run-time for bulk data transfer jobs, and it could provide "Application-aware" networking services. It was also able to support differentiated service to ensure the required QoS services for priority traffics. It provisioned the QoS-guaranteed network paths between two ends with the fastest possible time and it also provided the enterprise level network services along with the integration of ESNet OSCARS WAN connection services. Ideally, it is very import to compare the AmoebaNet solution with other available solutions with respect to the performance and capabilities. However, these comparisons are very difficult for us because of the following reasons. First, there are a limited number of solutions available to provide the dynamic network programmability feature for bulk data transfers in large-scale distributed scientific environments. Second, although there are many SDN research projects and efforts, very few projects have developed software packages or systems that can be practically deployed. In future work, we will work on the enhancements of current architecture of AmoebaNet. These features includes: (a) new primitives for advance reservation systems; (b) generalized scheduler that coordinates with multiple AmoebaNet's services located in different sites; (c) enablement of multi-domain functionality; (d) improvement of existing algorithms; and (e) per protocol and per flow based QoS path computation.

**Author Contributions:** Conceptualization, S.A.R.S.; methodology, S.A.R.S. and S.-Y.N.; software, S.A.R.S.; validation, S.A.R.S.; formal analysis, S.A.R.S.; resources, S.-Y.N.; writing—original draft preparation, S.A.R.S.; writing—review and editing, S.-Y.N.; supervision, S.-Y.N.; funding acquisition, S.-Y.N.



**Funding:** This work was supported by the National Research Foundation of Korea (NRF) grant funded by the Korea government (MSIT) (No. NRF-2008-00458).

**Acknowledgments:** This work has been performed while the authors were participating in the joint-project between Fermi National Accelerator Laboratory and Korea Institute of Science and Technology Information.

**Conflicts of Interest:** The authors declare no conflicts of interest.

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
