# Peer review of "A Dynamic Programmable Network for Large-Scale Scientific Data Transfer Using AmoebaNet"

_applsci, doi:10.3390/app9214541_

Round 1
Reviewer 1 Report
The problem researched by this paper is well motivated and the proposed method makes technically sense.
The evaluation cases are insufficient, for example experiments should be scaled to higher throughput with high bandwidth capabilities because the overall goal of this study is extreme scale. Moreover, based on the work of Liu. et al., doi.org/10.1109/CCGRID.2019.00023, doi.org/10.1145/3208040.3208053, doi.org/10.1145/3208040.3208053 (consider cite them), it is quite a different story when experimenting in high bandwidth environments.
Moreover, the proposed design should be evaluated using more transfer applications, e.g., the widely used GridFTP (or Globus) should be one of them (especially when you use ESnet because based on 10.1145/3208040.3208053, more than 50% of the traffics over ESnet are via GridFTP).
Figure 6, 7 and 8 do not give any valuable information but a waste of space, they should be removed or replotted the throughput in fig 8 if really necessary.
I noticed many typo and grammatical issues, please make a proof reading. e.g., the 2nd last sense of section 3 missed ‘is’.
Author Response
Response to Reviewer 1:
Reviewer’s comments are very helpful and very much appreciated. We are very much thankful to reviewers for their thoughtful comments and respond as follows.
Reviewer’s Point #1: The evaluation cases are insufficient, for example experiments should be scaled to higher throughput with high bandwidth capabilities because the overall goal of this study is extreme scale.
Response to Point #1: Thank you for suggesting us a new experiment. Our all evaluated test cases performed in real environments and with real software suits. Because of limitations of bandwidth availability at ESNet, we were not able to extend our experiments up to 40Gbps. However, to proof the working capability of AmoebaNet in extreme scale with high bandwidth, we included a new local experiment discussed in section 4 (sub-section 4.1).
Reviewer’s Point #2: Based on the work of Liu. et al., doi.org/10.1109/CCGRID.2019.00023, doi.org/10.1145/3208040.3208053 (consider cite them), it is quite a different story when experimenting in high bandwidth environments.
Response to Point #2: We extensively studied the reviewer’s suggested papers and included in our manuscript in related work section 3. However, these papers are out of our scope because dealing with small and large scale datasets is the responsibility of mdtmFTP (a module of BigData Express).
Point #3: The proposed design should be evaluated using more transfer applications, e.g., the widely used GridFTP (or Globus) should be one of them (especially when you use ESnet because based on 10.1145/3208040.3208053, more than 50% of the traffics over ESnet are via GridFTP)
Response to Point #3: Yes, we only evaluated our AmoebaNet solution with mdtmFTP which is working as a core data transfer software inside BigData Express (BDE) application. As per a study mdtmFTP is much better than GridFTP in performance, referred to results shown here: https://mdtm.fnal.gov/Evaluation.html. It is also possible to evaluate our solution with GridFTP application, but the additional software (such as network agent and distributed scheduler) will be required inside GridFTP which is not possible in this short period of time.
Point #4: Figure 6, 7 and 8 do not give any valuable information but a waste of space, they should be removed or replotted the throughput in fig 8 if really necessary.
Response to Point #4: As per reviewer’s suggestion, we removed the Figure 6 and 7. However, the Figure 8 cannot be replotted because it is directly generated by scientific application (BDE).
Point #5: I noticed many typo and grammatical issues, please make a proof reading. e.g., the 2nd last sense of section 3 missed ‘is’.
Response to Point #5: We did some extensive review of our writing and also tried to reduce some inaccuracies in our manuscript.

Reviewer 2 Report
This paper developed a dynamically programmable prototype network to facilitate convenient use and better performance for large-scale scientific applications to move their big datasets for distributed storage and processing. Such convenience was obtained by applying software-defined networking (SDN) technology to accordingly adapt the network with the applications that invoke the data transfer.
Overview, this paper is well-motivated and aims to tackle an important problem. But I have the following questions/comments that I feel should be addressed:
what “out of box solutions”, as mentioned at the end of page 1, are required for those data transfer tools to be able to meet the large-scale data transfer? adding some examples of such solutions would be better. end-to-end big data transfer requires the high-speed dedicated connections to be reserved in advance, such reservation involves different segments of a network path. While services like OSCARS and ION/AL2S can be used for connection reservation in the backbone part, services like AmoebaNet, the one being developed in this work, can be used for such purpose in campus and LAN, how about the regional/edge network segments? The paper claimed to “focused on an extension of AmoebaNet with some latest development of dynamic programmability”, however, I did not find any description about such latest developments but just some experimental case studies presented with screenshots There is a missing reference at the 4th line of the Introduction section; and at line 5 to line 5, it should be “…implemented a number of …”; following that the word “although” and “however” are in grammatical conflict. A careful proofread would be appreciated. The fonts of the words in Fig. 3 looks like messed upAuthor Response
Response to Reviewer 2:
Reviewer’s comments are very helpful and very much appreciated. We are very much thankful to reviewers for their thoughtful comments and respond as follows.
Reviewer’s Point #1: What “out of box solutions”, as mentioned at the end of page 1, are required for those data transfer tools to be able to meet the large-scale data transfer? Adding some examples of such solutions would be better.
Response to Point #1: In the context of “out of box solution” means a true end-to-end dedicated solution for large-scale data transfer. To consider the “out of box solution”, we claimed that the current networking paradigm must address the following major problems: scalability, last mile, and dynamic programmability problems. We tried to address these issues using AmoebaNet service.
Reviewer’s Point #2: End-to-end big data transfer requires the high-speed dedicated connections to be reserved in advance, such reservation involves different segments of a network path. While services like OSCARS and ION/AL2S can be used for connection reservation in the backbone part, services like AmoebaNet, the one being developed in this work, can be used for such purpose in campus and LAN, how about the regional/edge network segments?
Response to Point #2: Both AmoebaNet and OSCARS solutions are working together and provide the seamless end-to-end dedicated connections. All the regional/edge network segments connections dynamically established by using ESNet OSCARS service and from boarder router of each edge network to end DTNs is responsibility of AmoebaNet. As shown in section 4 (Figure 8), all the segments connectivity dynamically done by using OSCARS service and it provide dedicated 1Gbps connection from KISTI’s boarder router to FNAL boarder router. Whereas, remaining both campuses dedicated connections is the responsibility of AmoebaNet.
Point #3: The paper claimed to “focused on an extension of AmoebaNet with some latest development of dynamic programmability”, however, I did not find any description about such latest developments but just some experimental case studies presented with screenshots
Response to Point #3: We added new sub-section 2.2 which included the description about latest developments of dynamic programmability features of AmoebaNet. All these features supported by BigData Express (BDE) application to establish the end-to-end paths.
Point #4: There is a missing reference at the 4th line of the Introduction section; and at line 5 to line 5, it should be “…implemented a number of …”; following that the word “although” and “however” are in grammatical conflict. A careful proofread would be appreciated.
Response to Point #4: We did some extensive review of our writing and also tried to reduce some inaccuracies in our manuscript.
Round 2
Reviewer 1 Report
new version looks ok to me.
Author Response
We are again very much thankful to reviewer for his/her valuable feedback.
Reviewer 2 Report
Authors’ responses are appreciated and the paper’s quality has been improved more or less.
I feel that the paper should make it more clear about its contribution, as AmoebaNet alone does not provide any end-to-end big transfer solutions but merely establishes dedicated connections at the edge and campus segments of a network path using SDN-based solutions. Services like Esnet’s OSCARS and Internet’s AL2S/ION are equally important (if not more) as they must be used to reserved dedicated connections in backbones.
The paper needs significant editorial work as there are numerous typos as well as readability issues. To just name a few: i) Aren’t Figure 3 and deleted Figure 2 the same? What changes have been made?; ii) The beginning of page 11, the figure’s caption is missing; iii) Line166 “… an application can be enable or disable” … should it be “can enable or disable” or “can be enabled or disabled”?; and ...
Author Response
We are again very much thankful to reviewers for his/her valuable comments and suggestions. We would like to respond as follows.
Reviewer’s Point #1: I feel that the paper should make it more clear about its contribution, as AmoebaNet alone does not provide any end-to-end big transfer solutions but merely establishes dedicated connections at the edge and campus segments of a network path using SDN-based solutions. Services like Esnet’s OSCARS and Internet’s AL2S/ION are equally important (if not more) as they must be used to reserved dedicated connections in backbones.
Response to Point #1: The contribution of our paper is the extension of AmoebaNet and it is a true end-to-end network solution for large scale data transfer. However, because of limitation of deploying the AmoebaNet at core/backbone layer, we just tested our solution at edge along with the existing backbone solution (i.e. OSCARS) and showed that our solution is capable to support the programmability features. But, it doesn’t mean that our system is not capable of working alone in large system including core/backbone networks. At present, it is not possible to demonstrate AmoebaNet at core/backbone level because of some hardware and simulation limitations. Currently, we are working with scientific network providers (KROERNET & ESNet) and trying to setup AmoebaNet only network at both (edge and core), hopefully, in future we will be able to demonstrate this capability of AmoebaNet.
Reviewer’s Point #2: The paper needs significant editorial work as there are numerous typos as well as readability issues. To just name a few: i) Aren’t Figure 3 and deleted Figure 2 the same? What changes have been made?; ii) The beginning of page 11, the figure’s caption is missing; iii) Line166 “… an application can be enable or disable” … should it be “can enable or disable” or “can be enabled or disabled”?; and ...
Response to Point #2: We did an extensive editorial work for our paper and also tried to reduce some inaccuracies. Hopefully, this time it will be give you a good picture. Yes deleted the Figure 3 is the same, however, we just change its number because we added new Figure 2 in reviewed manuscript. Please check again the manuscript, I can see the caption of Figure 5 on page 11. Finally, we tried our best to remove the typos as per your given suggestions.